# Portable Prussian Blue-Based Sensor for Bacterial Detection in Urine

**DOI:** 10.3390/s23010388

**Published:** 2022-12-30

**Authors:** Carolin Psotta, Vivek Chaturvedi, Juan F. Gonzalez-Martinez, Javier Sotres, Magnus Falk

**Affiliations:** 1Department of Biomedical Science, Faculty of Health and Society, Malmö University, 20506 Malmö, Sweden; 2Aptusens AB, 29394 Kyrkhult, Sweden; 3Biofilms-Research Center for Biointerfaces, Malmö University, 20506 Malmö, Sweden

**Keywords:** portable sensing, bacterial detection, Prussian blue, urine analysis

## Abstract

Bacterial infections can affect the skin, lungs, blood, and brain, and are among the leading causes of mortality globally. Early infection detection is critical in diagnosis and treatment but is a time- and work-consuming process taking several days, creating a hitherto unmet need to develop simple, rapid, and accurate methods for bacterial detection at the point of care. The most frequent type of bacterial infection is infection of the urinary tract. Here, we present a wireless-enabled, portable, potentiometric sensor for *E. coli*. *E. coli* was chosen as a model bacterium since it is the most common cause of urinary tract infections. The sensing principle is based on reduction of Prussian blue by the metabolic activity of the bacteria, detected by monitoring the potential of the sensor, transferring the sensor signal via Bluetooth, and recording the output on a laptop or a mobile phone. In sensing of bacteria in an artificial urine medium, *E. coli* was detected in ~4 h (237 ± 19 min; n = 4) and in less than 0.5 h (21 ± 7 min, n = 3) using initial *E. coli* concentrations of ~10^3^ and 10^5^ cells mL^−1^, respectively, which is under or on the limit for classification of a urinary tract infection. Detection of *E. coli* was also demonstrated in authentic urine samples with bacteria concentration as low as 10^4^ cells mL^−1^, with a similar response recorded between urine samples collected from different volunteers as well as from morning and afternoon urine samples.

## 1. Introduction

Bacterial diseases constitute some of the leading causes of mortality globally, and early infection detection is critical regarding diagnosis and treatment [1]. One of the most frequently occurring bacterial infections is urinary tract infection (UTI) [2]. The majority of UTIs are caused by *Escherichia coli* (*E. coli*) of enteric origin, but UTI can also be caused by *Enterococcus* and *Klebsiella* species [3,4,5,6]. UTI can be classified into uncomplicated and complicated cases and can be accompanied by various symptoms related to the severity of the infection [3,4,5]. Symptoms can range from mild, such as fever and nausea, to severe, such as circulatory infection and organ failure in extreme cases. With unhindered bacterial growth and no intervening treatment, the colonialization starts from the urethra (urethritis), can go further up to the urinary bladder (cystitis), kidney (pyelonephritis), and in severe cases, into the bloodstream (sepsis) [3,6]. Diagnosis of UTI can be established from clinical symptoms, with subsequent laboratory test confirmation. The most common approaches to pathogen detection and quantification are based on cell culture methods taking 48 to 72 h, which is also the gold standard to diagnose UTI in a clinical context [2,3]. In general, healthy urine should be sterile [7], and therefore, a urine culture with 10^4^ to 10^5^ CFU mL^−1^ or above confirms UTI [2,3,8]. The diagnosis of UTI is challenging since it is a time- and work-consuming process taking several days to establish a diagnosis, creating a hitherto unmet need to develop simple, rapid, and accurate methods for detection at the point of care (POC) [2,3,7,9]. This is particularly problematic in elderly patients, most of whom are asymptomatic, where early diagnosis and treatment are crucial because even a brief delay contributes to mortality [10].

Enabled by the ongoing miniaturization of electronics, new portable instruments combining sensing with wireless communication can be designed, which minimizes the need for bulky laboratory infrastructure at the POC and primary care settings [11].

The use of POC devices allows decentralized clinical monitoring and analysis, which decrease associated costs and provide an improvement in patient health quality. Electrochemical sensors are attractive to use in POC settings since they typically offer high sensitivity and selectivity, low costs, simple operation, portability, and rapid response time. A fast, low-cost, user-friendly, and easy-to-use detection system for bacteria in real-time would help to mitigate the limitations of current detection methods, providing a critical tool in combating infections by allowing for faster and more effective therapeutic intervention. Moreover, the management of UTI via POC devices can avoid improper antibiotic prescription and therefore could help decelerate the antibiotic resistance problem [12]. Furthermore, deploying artificial intelligence and cloud-computing to data recorded by POC devices can provide the basis for next-generation smart health care devices and have, for example, been used in POC optical based sensors for blood cell and coagulation analysis as well as for analyzing urine and stool [13,14,15]. Different machine learning algorithms are also powerful tools for analyzing multiplexed POC sensing for making health assessments, for instance, by combining different electrochemical sensors, as health analysis typically is complex and single-marker analysis is insufficient [16].

Electrochemical sensors present a rapid bacterial detection method, where most sensors rely on amperometry or impedance measurements, typically based on interactions with antibodies, aptamers, or antimicrobial peptides [1]. Furthermore, screen printing of electrodes is conducive to lowering cost and improving biosensor reliability, with a variety of commercially available screen-printed electrodes (SPEs) now available, enabling the design of cost-efficient disposable sensing systems. However, while electrochemical systems for bacterial detection can achieve very high sensitivity, they can become rather complex regarding sensor design, measurement protocols, and digital decoding of the output signal from the sensor. Ideally, a sensor should be inexpensive, small, and easy to operate with little or no sample preparation.

Prussian blue (PrBl) is a well-known electrocatalyst, where PrBl films can be electrochemically deposited and have been used to design a wide variety of electrochemical sensors [17,18]. PrBl, formed from inexpensive and abundant metals, is attractive due to its low toxicity and relatively simple and economical synthesis processes [17,19]. PrBl in various forms has been shown to possess peroxidase-like and catalase-like activity, and PrBl nanoparticles have also been employed as effective scavengers of reactive oxygen species [20,21]. PrBl has recently been used in the design of a variety of portable or wearable analytical devices, all of which were further modified with enzymes, and enzymatic reduction of PrBl to Prussian white (PrWh) was detected amperometrically [22,23,24]. Furthermore, ferricyanide is a common metabolic indicator for both Gram-positive and Gram-negative bacteria, and recent studies have shown that PrBl can be reduced to PrWh through the metabolic activity of a variety of microorganisms, including *E. coli*, a feature that has been used to design electrochromic sensors for bacteria [25,26,27,28].

The possibility to detect bacteria in aqueous environments using a low-cost technology could greatly simplify management of UTIs, but also be of interest for several other applications. For example, bacterial infections of wounds have many serious implications, where infection can result in conditions like erythema, edema, and necrosis. In addition to being used to analyze body fluids, bacterial detection is important in a wide range of other applications, such analyzing the quality of food and drinking water.

Here, we present a portable potentiometric sensor for *E. coli*. Although potentiometry is widely used in the biosensor field, there are only a few examples of potentiometric biosensors for detecting whole bacterial cells [1,29,30]. The sensing principle is based on the electrochemical reduction of PrBl to PrWh triggered by the metabolic activity of the bacteria. The sensor can be easily assembled, and a prototype was designed based on PrBl-modified screen-printed electrodes (SPEs). The sensor output was wirelessly transferred via Bluetooth and was recorded on a laptop or a mobile phone. Sensing of bacteria was demonstrated both in a complex buffer as well as in human urine samples. Instead of using the traditional time-consuming procedures for detecting and identifying bacteria, the current detection system offers a cost-effective method that can potentially be used to detect infected urine samples using an inexpensive disposable sensor in a POC setting without the need for a specialist user.

## 2. Materials and Methods

### 2.1. Chemicals

To mimic urine, a complex liquid mixture was prepared according to a previously established protocol [31]. The artificial urine medium (AUM) contained various components, listed in Table 1, and was adjusted to pH 6.4 with NaOH or HCl. Magnesium sulfate was purchased from Kebo Lab AB (Stockholm, Sweden). Potassium dihydrogen phosphate was purchased from Duchefa Biochemie (Haarlem, The Netherlands). All other chemicals were purchased from Merck (Darmstadt, Germany). All solutions were prepared with MilliQ-water, 18.2 MΩ cm^−1^.

Mid-stream human urine samples were collected from apparently healthy volunteers. One sample was collected once in the afternoon (non-fasting); the sample was further divided and either stored in the refrigerator for use on the next day or frozen for later use, ensuring that the same urine sample was used for the initial investigations in urine. In addition, a morning sample was collected from the same volunteer and a morning and an afternoon sample from an additional volunteer, treated in the same way.

### 2.2. Preparation and Characterization of PrBl-Modified SPEs

Disposable screen-printed 220 AT gold electrodes were purchased from Metrohm Dropsens (Oviedo, Spain). The SPEs were first carefully rinsed with Milli-Q water, and electrochemical modification and characterization were performed using a μAutolab Type III/FRA2 potentiostat/galvanostat from Metrohm Autolab B.V. (Utrecht, The Netherlands). A standard three-electrode configuration was used with an Ag/AgCl (3 M KCl) reference electrode and a platinum counter electrode. PrBl synthesis was carried out in 0.1 M hydrochloric acid containing 0.1 M potassium chloride and an equimolar mixture (10 mM each) of FeCl_3_ and K_3_Fe (CN)_6_. Electrodeposition of PrBl was achieved by applying a constant potential of 0.4 V vs. Ag/AgCl (3 M KCl) for 10, 40, or 100 s, respectively. Electrodes were then carefully washed with Milli-Q water and transferred to a supporting electrolyte solution (0.1 M KCl, 0.1 M HCl) and activated by cycling the potential between 400 mV and –100 mV for 10 cycles at a sweep rate of 100 mVs^−1^.

### 2.3. Wireless Sensing

The wireless prototype to measure open circuit potential (OCP) was comprised of an Adafruit HUZZAH32—ESP32 Feather, an ADS1115 (16 bit) digital to analog converter, and a 3.7 V 1400mAH Lipo battery (Appendix A). The Adafruit HUZZAH32—ESP32 is a user-programmable microcontroller with RFID and Bluetooth connectivity. The Arduino version 1.8.19 software was used to code and program the Adafruit HUZZAH32—ESP32 Feather board connected to ADS1115 (16 bit) digital to analog converter; the conversion rate was 860 samples per second. The microcontroller was set to measure the OCP once every five minutes, and the measured value was the averaged result of the 860 samples taken by the ADS1115. To disconnect the sensor from the measurement circuit between measurements, a relay switch was coupled with the wireless sensing system. The continuous data supplied by the Adafruit HUZZAH32 were collected using a smartphone or laptop. To verify the accuracy of the data recorded by the Arduino, the measured OCP values were compared with the values obtained with a regular multimeter, confirming the validity of the data collected by the Arduino.

Wireless measurements of the OCP in AUM and human urine were performed using an external Ag/AgCl (3 M KCl) reference electrode. The continuously stirred and sealed beaker containing AUM or human urine was connected to a water bath to maintain a constant temperature of 37 °C. KCl solution was added to the urine sample (final concentration of 0.1 M) to minimize the effect of different KCl concentrations on the recorded electrode potential. Before the addition of bacteria, a baseline (~30 min) was recorded to ensure the stability of the PrBl electrode in the medium. Afterward, the desired volume of the bacteria aliquot was added. The measurement was run continuously for at least 24 h.

### 2.4. Bacteria Cultivation

To inoculate single bacterial colonies, *E. coli* (ATCC 2592) was grown on sterile lysogeny broth (LB) agar plates. One colony *E. coli* was then taken and grown overnight in 40 mL AUM or urine in the incubator at 37 °C.

Before each measurement, quantitative bacterial analysis was performed with a spectrophotometer (Pharmaspec UV-1700 spectrophotometer, Shimadzu, Kyoto, Japan) by measuring the transmission at 600 nm. The optical density measurements were used to estimate the amount of cells per milliliter (OD_600_ = 1.0 = 8 × 10^8^ cells/mL; [32]). An aliquot was taken from the overnight culture in the appropriate medium when the bacteria were in the lag-phase of the growth curve with an optical density of OD_600_~0.1 (Appendix A) to avoid large variations in the experimental starting conditions and added aliquot volumes.

### 2.5. Flow Cytometry

To monitor the growth of *E. coli* in solution during the measurement, flow cytometry was performed using a dsDNA quantitation kit (Pico488 from Lumiprobe GmbH, Hannover, Germany): 1 mL of undiluted AUM was collected at different time intervals from the beaker used for sensing and centrifuged at 5000 rpm (2000× *g*), 5 min at room temperature. Then, 950 µL of the supernatant was removed, and the remaining solution was resuspended in 150 µL 1:200 SYBR Green (Pico488 from Lumiprobe GmbH, Hannover, Germany) and 150 µL staining buffer (phosphate-buffered saline/0.01% Tween−20/1 mmol ethylenediaminetetraacetic acid), incubated for 10 min at 37 °C. Next, a fixed volume (25 µL) of stained bacteria was analyzed with the flow cytometer (BD Accuri ™ CG plus from Becton Dickinson, Franklin Lakes, NJ, USA) using a low sample rate and a selected threshold setting on FSC (forward-scatter) and SSC (side-scatter). An unstained sample was used to gate the bacteria since SYBR Green only binds to DNA and no other compounds in the artificial urine medium. The amount of SYBR Green stained bacteria/cells per mL could be calculated by multiplying the number of stained bacteria detected in the dot plot by the conversion factor of 16 (Appendix A). Aliquots for the analysis were taken at the specific incubation times of 0, 2, 4, and 6 h (Appendix A). The protocol for counting bacteria in AUM was based on a previously described method by Moshaver et al. [33]. The protocol was tested prior to measurement in AUM and human urine, with different dilutions to ensure a linear dependency of the measured cell concentration (Appendix A).

### 2.6. Scanning Electron Microscopy

Scanning electron microscopy (SEM) was performed on PrBl-modifed SPEs, sputtered with gold using an Agar automatic sputter coater at 30 mA, 0.08 mbar pressure, and with a sputtering time of 40 s. SEM images were obtained using a Zeiss EVO LS10 scanning electron microscope (Carl Zeiss NTS Gmbh, Standort Göttingen, Germany) equipped with a LaB6 filament.

## 3. Results and Discussion

### 3.1. Electrodeposition and Characterization of PrBl-Modified SPE

Three different times for potentiostatic electrodeposition were investigated, specifically, 10, 40, and 100 s, and the corresponding cyclic voltammograms (CVs) at different times are shown in Figure 1a. The CVs are typical for PrBl-modified electrodes, undergoing the reversible redox reaction from colored PrBl to the reduced colorless form, PrWh, where the entrapment of cations compensates the transfer of electrons in the film according to the equation:(1)$$Fe_{4}^{III}{\left[Fe^{II}{\left(CN\right)}_{6}\right]}_{3}+4e^{-}+4K^{+}↔K_{4}Fe_{4}^{II}{\left[Fe^{II}{\left(CN\right)}_{6}\right]}_{3}$$

Longer electrodeposition times produced thicker PrBl films with deeper color intensities and larger current magnitudes in the CVs. After 10 s deposition, the color change was barely visible, but distinctly blue deposits were observed after longer deposition times. In addition, peak separation also increased with the electrodeposition time (41, 81, and 96 mV for 10, 40, and 100 s, respectively). The coverage of PrBl on the electrode surfaces was determined by integrating the cathodic currents, corresponding to the reduction of PrBl to PrWh. Charge densities were calculated using the electrode geometric area for the different deposition times. Although the actual surface areas of the SPEs were not determined, the charge density analysis of the peaks showed that 40 and 100 s deposits contained roughly 2.4 and 3.5 times more PrBl than the amount after 10 s electrodeposition.

The final time for deposition was fixed to 40 s for further experiments, as shorter deposition times produced very thin layers, and longer times resulted in CVs with higher capacitive currents and ∆Ep values close to 100 mV. Thus, 40 s was chosen as a compromise, since the 40 s deposition produced a comparatively much thicker, distinctly blue layer than 10 s deposition, with peak currents increasing almost linearly with increasing scan rate (Figure 1a, inset). Figure 1b shows a typical SEM image of an SPE electrodeposited at 0.4 V for 40 s. Recent studies have shown a strong dependence on the morphology/growth orientation of the samples as a function of applied potential and thickness, together with the formation of cracks, as also observed here [34].

Following electrodeposition and subsequent activation of the electrodes, dried modified electrodes could be stored for several weeks (the longest time investigated was 4 weeks) without any significant deterioration. After three days of storage in AUM, the amount of PrBl on the surface had decreased by roughly thirty percent, judged by integrating the cathodic current in 0.1 M HCl and 0.1 M KCl and comparing the results before and after incubation in AUM.

### 3.2. Wireless Sensing Principle Using PrBl-Modified SPEs

Most studies using PrBl-modified electrodes rely on amperometric detection. In a recent study on *E. coli* detection, the bacteria slowly reduced PrBl to PrWh, and longer incubation times produced higher currents when an oxidative potential was applied [28]. During amperometric detection, the reduced PrWh is reoxidized to PrBl. To allow for continuous monitoring, we instead used potentiometric detection. The potential of the sensors is determined by the activity of potassium ions and the ratio of the oxidized and reduced form of the PrBl film, according to the following equation:(2)$$\text{E}={\text{E}}^{0}+\frac{\text{RT}}{\text{F}}\ln\frac{\text{PrBl}\times \text{K}}{\text{PrWh}}$$
where PrBl and PrWh denote activities of the individual redox forms, and K refers to the activity of potassium ion in the solution phase adjacent to the film [35]. The mid-peak potential is a function of potassium ion activity, and with decreasing electrolyte concentration the equilibrium potentials are displaced toward more negative values. The variations of the electrode potentials measured potentiometrically at a fixed ratio of oxidized/reduced film have been used for designing ion-sensitive electrodes [35,36]. Upon investigation of the modified SPEs used, very close to a 59 mV shift per decade of KCl concentration was observed (Appendix A). Similarly, decreasing the degree of oxidation of the iron hexacyanoferrate matrix will shift the electrode potential to more negative values. Simplistically, the OCP is thus determined by the redox state of the film with the Donnan potential added, and by monitoring the change in electrode potential versus a suitable reference electrode, a potential decrease can be related to bacterial activity [36].

To wirelessly monitor the potential of a PrBl-modified SPE, a sensing platform with wireless capabilities was designed, as illustrated in Figure 2. Briefly, in the presence of bacteria PrBl is reduced to PrWh and the OCP of the sensor is continuously monitored. An Arduino-based microcontroller with Bluetooth connectivity was used to monitor the OCP of the PrBl-modified SPE vs. reference, transmitting the data continuously to an external device. Upon reduction of PrBl, a negative shift was observed in the recorded potential, and eventually the electrode changed color from blue-green to gold, as PrBl was reduced to PrWh.

### 3.3. Wireless Sensing in AUM and Human Urine

The developed wireless-enabled sensing system was first used to detect *E. coli* in AUM by adding different numbers of pre-incubated bacteria while monitoring the OCP. After 30 min in AUM, either a 2 µL or 200 µL aliquot of bacteria was added to the AUM solution (volume 60 mL). A typical response is shown in Figure 3a, where a 2 µL aliquot corresponded to roughly 10^3^ cells mL^−1^. The precise OCP values measured at selected regular intervals are also listed in Table 2. The number of cells added were estimated based on an optical density measurement of the bacterial suspension pre-incubated overnight. It should be noted that this estimate is not precise and also does not necessarily correspond to living cells, contributing to variations in the OCP measurements [37]. As control measurements, the stability of PrBl-modified SPEs were measured in AUM without added bacteria, as well as monitoring the change in OCP of an unmodified SPE on exposure to bacteria.

The OCP of the PrBl-modified electrode in AUM without bacteria dropped initially from ~140 mV but became stable over the next hours with an average OCP of 92 mV, ranging from ~89 to 95 mV. A similar OCP development was initially observed following the addition of 2 µL of a bacterial aliquot (solid line, Figure 3a). The similarity of the initial response of the bacteria-free AUM and AUM with bacteria added can be related to the fact that the bacteria acclimatize and adjust to the surroundings before changing their metabolism and transitioning from the lag-phase to the exponential growth phase [38]. After a few hours, a clear negative shift of the OCP is observed. Based on the OCP variation observed in native AUM (6 mV difference between the highest and lowest baseline values), taking a cutoff of two times this variation as a detection limit for *E. coli*, bacteria could be detected after 223 min ± 22 min (n = 6) from a starting concentration of ~10^3^ cells mL^−1^. After 200 µL of bacterial aliquot was added, an almost immediate sensor response was observed, with bacteria being detected within 21 ± min 7 min (n = 3) from a starting concentration of ~10^5^ cells mL^−1^. The rapid response at higher concentrations can in part be attributed to the higher aliquot volume, since reductive metabolites excreted during the overnight culture growth will be present at larger concentrations [38,39,40]. Previous studies have shown that upon removal of evolved *E. coli* cells, the growth medium was redox active, attributable to, for example, c-type cytochromes or quinones secreted by bacteria [41].

In order to relate the negative shift in OCP observed to the growth of bacteria in the sample in more detail, flow cytometry was performed at regular time intervals (0, 2, 4 and 6 h) after the addition of a 2 µL aliquot of lag-phase bacteria, (Appendix A), while also measuring the OCP. The results are shown in Figure 3b and tabulated in Table 2. In this way, the actual number of living bacteria was monitored in parallel with OCP measurements. Based on flow cytometry, the starting cell count was ~7000 cells mL^−1^. The *E. coli* lag-phase in AUM can last for up to 3 h depending on the starting cell count and the state of growth of the bacteria. A slow initial growth was observed, and the number of bacteria doubled in 2 h (~15,700 cells mL^−1^). The lag-phase is defined by metabolically active bacteria disengaged in active/rapid cell division, but over time bacteria adapt to the new growth conditions [38,42]. After the lag-phase, rapid growth was observed, with ~240,000 cells mL^−1^ after 4 h and ~4,160,000 cells mL^−1^ after 6 h. The growth phase was still ongoing when the last measurement point after 6 h was taken (Appendix A). The start of the bacterial growth phase, characterized by increased metabolic activity of the bacteria, correlated very well with the observed negative shift in OCP. In contrast, the measurement of a PrBl-modified electrode in native AUM showed, besides the aforementioned initial drop, no change in OCP over 24 h (Appendix A). Thus, the results from native AUM confirmed that the negative OCP shift should be attributed to the actions of *E. coli*. The OCP for an unmodified SPE in AUM changed during the different growth phases and varied from 28 mV up to 82 mV. Nevertheless, no correlation between the OCP values and the growth of bacteria was observed. This shows the importance of the PrBl-modification to obtain a rapid sensor response to bacteria.

The growth phase in AUM lasts up to 9 h (Appendix A) and with longer incubation times, the OCP fluctuated (Appendix A). Following the growth phase, the bacteria enter into a stationary phase, characterized by balanced cell death and cell division. However, with increasing incubation time, nutrients become less available, and the level of toxins increases [42]. Besides the changing composition of electroactive species in solution, the fluctuations over time could have been caused by bacterial adsorption on the electrode surface itself, and *E. coli* have been shown to slowly deposit on surfaces, especially in the stationary phase [39,43,44]. This makes interpretation of the further observed change in OCP difficult without further analysis of changes occurring at the electrode surface and in solution. However, this is beyond the scope of this study, as it does not affect the initial detection of bacteria.

In addition to AUM, bacterial detection was also accomplished in samples of human urine, with aliquots of 10 µL, 100 µL, and 200 µL of a lag-phase bacterial suspension added to the solution after about 30 min (using the same urine collected in the afternoon, which was then divided into different test samples). A typical response is shown in Figure 4a, where a 10 µL aliquot corresponded to roughly 10^4^ cells mL^−1^. When no bacteria were added to the urine a similar response as in AUM was observed, that is, the OCP initially decreased and then stabilized around slightly below 90 mV. However, in contrast to the measurements in AUM, a positive OCP shift was initially observed when bacteria were added; the larger the aliquot added, the more pronounced the shift. To further investigate the detection of bacteria in urine, the response of the sensor was recorded in samples collected at different times as well as from a different volunteer, where 200 µL of a lag-phase bacterial suspension was added after about 30 min. The results are shown in Figure 4b, with a similar response observed from urine collected at different times as well as from different volunteers. The recorded OCP is dependent on the activity of cations small enough to enter the crystal lattice of PrBl, leading to a certain variation between samples to be expected. The observed positive shift upon addition of bacteria could be related to different metabolic activity of lag-phase bacteria in AUM compared to authentic urine, for example, causing a change in the activity of cations in the urine samples when bacteria was added. The metabolic activity of bacteria is very complex and poorly understood; however, a change from the lag-phase, growth phase, and the transition to the stationary phase is known to cause a shift in gene expression and metabolic pathways due to changing surrounding conditions for different bacterial species [38]. For example, under certain conditions *E. coli* is known to produce ammonia, which is then protonated to ammonium ion [45]. Since the hydrated ammonium ion is small enough to enter the PrBl lattice, analogous to the influence of potassium ions, the electrode potential is affected by the ammonium ion activity at the solid–liquid interface [46].

However, the detection times for bacteria in urine were similar to those in AUM, and the initial positive OCP shift was the only significant difference. Hence, as observed in AUM, when the bacteria entered the exponential growth phase (~3 h,) a rapid negative OCP shift was observed, with negative slopes similar to those found in AUM, and also, when entering the stationary phase, large OCP fluctuations was observed (Appendix A).

When demonstrating the sensing principle, the initial bacterial cell count as well as the aliquot growth phase need to be well defined. In all experiments, the bacterial aliquot was taken from the lag-phase to cover and investigate all bacterial growth phases in relation to the sensor response. The difference between the AUM and human urine measurements highlights the importance of measurements in physiological fluids since the complexity of the authentic fluids is difficult to reproduce in an artificial mixture. Furthermore, in physiological samples the composition of bacterial nutrients differs for each individual [47]. Further studies should thus be performed using urine from clinical samples, taken at different times of the day, to fully demonstrate the utility of the sensor as a rapid detection method for UTI. Furthermore, while a vast majority of UTIs are caused by *E. coli*, other bacterial species could be involved. Thus, since ferricyanide is a standard metabolic indicator for both Gram-positive and Gram-negative bacteria, the sensor response should also be investigated using other bacterial species besides *E. coli* known to cause UTI, such as Enterococcus and Klebsiella species.

A rather large variation was observed between different PrBl-modified SPEs regarding the time needed to reach a stable potential in both AUM and human urine. Depending on the particular PrBl modification, stabilization was almost immediate or could be delayed by over an hour. Also, fluctuations of the potential when approaching a stable baseline were modification dependent. Urine and AUM are very complex fluids, and it should be noted that the simplified Nernst equation, Equation (2), is not generally applicable for these complex systems where multiple reactions can take place. Nevertheless, to improve the sensor response, we will explore lower deposition potentials in order to produce smoother surfaces without cracks (see Figure 1b), as well as different deposition times. In addition, to reduce differences in the recorded OCP between samples, sufficient amounts of potassium could be added before each measurement, while making sure not to negatively affect the bacteria.

Currently, while the system in a fast and inexpensive way can show if a sample is infected or not, it is limited to only a rough estimate of the actual *E. coli* concentration, as while the time for the sensor to be reduced is correlated to the bacterial concentration variations were observed between different urine samples. By sending the senor signal to a mobile phone application, for example, where taking a sufficiently low recorded OCP as proof of bacterial infection, the application could then advise the user to seek further medical attention if such a potential were recorded by the sensor, otherwise conclude that no infection was detected. This is sufficient for the specific application of UTI detection, as traditionally any bacteria concentration of above 10^4^ to 10^5^ cells mL^−1^ would require further investigation by medical professionals, where the severity and the risk of the UTI mainly lays in the ascending infection due to colonialization of the bladder and further in the body and is not directly related to the concentration of bacteria. However, for many other applications the actual concentration may be of great interest, requiring further optimization of the system. To achieve low variability in OCP and response time to bacteria in different urine samples, sensors could be optimized by performing an experiment in which deposition time and potential as well as addition of potassium ions are varied. In this way, it is possible that the response time could be used to accurately predict the bacterial concentration, without the need of calibrating the sensors. For this, incorporating additional sensors may also be crucial, for instance, measuring the pH, nitrites, bilirubin or other important properties, analyzing the combined response using appropriate statistical models, which potentially also could enable differentiation between bacterial species.

## 4. Conclusions

Here, we demonstrate a novel use of PrBl-modified electrodes combined with a simple measurement circuit for measuring OCP to achieve detection of *E. coli*. The sensing principle was based on reduction of PrBl in the presence of bacteria, thus changing the potential of the sensing electrode. By using PrBl-modified SPEs and monitoring changes in OCP, detection of *E. coli* was achieved in several hours or faster, from initial cell counts ranging between 10^3^ and 10^5^ cells mL^−1^, which is the relevant bacterial range to be able to detect a UTI. Sensing was demonstrated in artificial urine as well as authentic urine samples from different volunteers collected both in the morning and in the afternoon. Only minor differences were observed between the different urine samples when the bacterial concentration was kept the same, thus demonstrating the applicability of the developed sensor in authentic physiological samples. The advantages of the proposed sensor concept include real-time, accurate, and highly sensitive detection of bacteria. Sensing was label independent and reagent-less. The sensor was easily assembled, based on inexpensive SPE, including a simple measurement circuit to record and wirelessly send the data, thus enabling low-cost scalable production and ease of use for the end-user. This demonstrates the potential for future application in POC or home analysis, eliminating the long time otherwise needed for sample collection, transport, preparation, and analysis (48–72 h). Future work will focus on improving the sensor reproducibility by further optimizing the PrBl deposition as well as expanding the investigation to other relevant bacterial species. Furthermore, to investigate calibrationless determination of the bacterial amount as well as differentiation of different bacterial species, inclusion of additional sensors combined with the use of appropriate statistical models will also be explored.

## Figures and Tables

**Figure 1 sensors-23-00388-f001:**
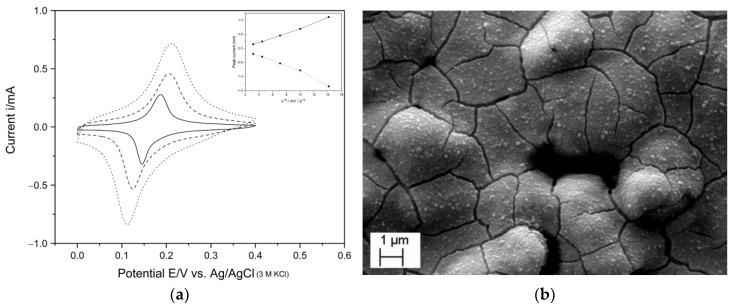
(**a**) CVs of PrBl-modified SPEs in 0.1 M HCl and 0.1 M KCl recorded for three different deposition times of 10, 40, and 100 s (solid = 10 s; dashed = 40 s; dotted = 100 s), recorded at a sweep rate of 50 mVs^−1^. Inset: anodic (solid) and cathodic (dotted) peak current vs. the square root of the scan rate; (**b**) SEM image of an SPE modified for 40 s with PrBl.

**Figure 2 sensors-23-00388-f002:**
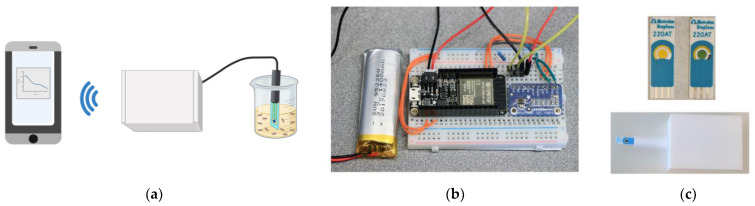
(**a**) Schematic representation of the wireless-enabled sensing principle, where the *E. coli* dependent potential of a PrBl-modified SPE is continuously measured by the Arduino, and the data sent to a suitable device (created with BioRender.com); (**b**) Photo of the wireless Arduino microcontroller; (**c**) Photographs of PrBl-modified SPEs in reduced (right) and oxidized (left) form together with an assembled wireless sensing platform, where the Arduino microcontroller is encased in a 3D-printed box and connected to an SPE for insertion in the test medium.

**Figure 3 sensors-23-00388-f003:**
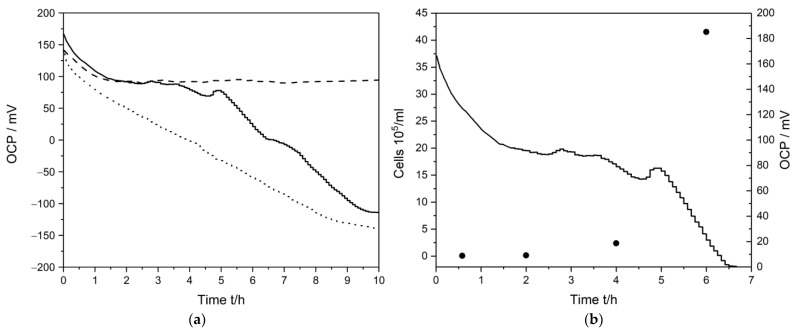
(**a**) Measurement of bacteria in AUM in which 2 µL (solid); 200 µL (dotted) or no (dashed) bacterial aliquot was added after 30 min; (**b**) Quantification of the bacteria with flow cytometry (solid circles, left *y*-axis) at 0, 2, 4, 6 h in relation to the measured OCP upon addition of a 2 µL bacterial aliquot (right *y*-axis).

**Figure 4 sensors-23-00388-f004:**
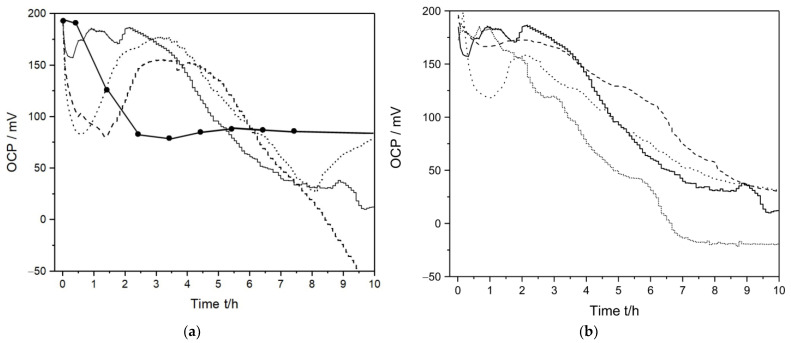
(**a**) Bacteria in human urine (volunteer one and afternoon collection), 200 µL (solid); 20 µL (dotted), 10 µL (dashed) or no (solid with circles) of a bacterial aliquot was added, (**b**) 200 µL of bacterial aliquots was added to different urine samples gathered at different times of the day and from two different volunteers, volunteer one and afternoon urine (solid), volunteer one and morning urine (dashed), volunteer two and afternoon urine (short dots), volunteer two and morning urine (dotted).

**Table 1 sensors-23-00388-t001:** Composition of urine-mimicking AUM, adjusted to pH 6.4, adapted with permission from Ref. [31] 2003, John Wiley and Sons.

Compound	Quantity (g)
Peptone L37	1
Yeast extract	0.005
Lactic acid	0.1
Citric acid	0.4
Sodium bicarbonate	2.1
Urea	10
Uric acid	0.07
Creatinine	0.8
Calcium chloride	0.37
Sodium chloride	5.2
Iron II sulphate	0.0012
Magnesium sulphate	0.49
Sodium sulphate	3.2
Potassium dihydrogen phosphate	0.95
Potassium hydrogen phosphate	1.2
Ammonium chloride	1.3
Distilled water	to 1 L

**Table 2 sensors-23-00388-t002:** Incubation time vs. OCP in AUM, comparing PrBl-modified and unmodified SPEs at different concentrations of bacteria.

Incubation Time (h)	OCP Unmodified SPE in 200 µL Bacteria (mV)	OCP PrBl SPE in 200 µL Bacteria (mV)	OCP PrBl SPE No Bacteria (mV)	OCP PrBl SPE in 2 µL Bacteria (mV)	Bacteria Concentration in 2 µL (Flow Cytometry) (Cells/mL)
0	28	141	141	125	7 × 10^3^
1	88	80	101	109	
2	72	51	92	92	1.5 × 10^4^
3	79	24	94	91	
4	82	−1	92	80	2.4 × 10^5^
5	57	−31	93	78	
6	40	−59	94	26	4.2 × 10^6^
7	57	−86	90	−6	
23	61	−168	99	−41	

## Data Availability

Most data is contained within the article. Additional data presented in this study are available on request to the corresponding author.

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
