# Peer review of "Portable Prussian Blue-Based Sensor for Bacterial Detection in Urine"

_sensors, 2022, doi:10.3390/s23010388_

Round 1

Reviewer 1 Report

Review comments

It is widely known that urinary tract infection refers to some diseases caused by the infection of patients' urethral orifice due to other factors. The specific source of urinary tract infection is often caused by bacterial infection, and only a few patients may be caused by virus or other protozoa infection.

In this study, the author focuses on an interesting topic, which aims at developing a portable prussian Blue-based sensor for bacterial detection in urine. The research design is reasonable and the experimental data are authentic, but the research has the following main problems.

1. As the author mentioned in the text as follows, the stability of the electrode needs to be improved.

Following electrode position and subsequent activation of the electrodes, dried modifiedelectrodes could be stored for several weeks (the longest time investigated was 2 weeks without any significant deterioration. After three days of storage in AUM, the amount of Prbl on the surface had decreased by roughly thirty percent , judged by integrating the cathodic current in 0.1 M HCl and 0.1 M KCl and comparing the results before and afterincubation in AUM.

2. In Table 2, the author has set the incubation time of 0, 1, 2, 3, 4, 5, 6, 7 and 23. Please explain the basis for setting the time points.

3. It can be seen from Table 2 that the electrode performance is not particularly stable.

4. In the article, the author mentioned that However, in contrast to the measurements in AUM , a positive OCP shift is initially observed when bacteria are added ; the larger the aliquot added , the more pronounced the shift . This could be related to different metabolic activity off lag-phase bacteria in AUM compared to authentic urine. According to the working principle of the electrode, this explanation seems weak. 

5. This study gives people a sense of "fine start and poor finish", which is the most obvious disadvantage of this study. In fact, researchers did not pay more attention to the detection of real urine samples. In fact, the researchers can also be aware of this problem. It is suggested to improve this part, as the authors mentioned in the paper -

Further studies should thus be performed using urine from many different donors, taken at different times of the day, to fully demonstrate the utility of the sensor as a rapid detection method for UTL. In addition, since ferricyanide is a standard metabolic indicator for both Gram-positive and Gram-negative bacteria, the sensor response should also be investigated using other bacterial species besides E . coli known to cause UTL, such as Enterococcus and Klebsiella species.

Reviewer 2 Report

The manuscript introduces a portable, fast and accurate method for bacterial concentration detection. Using bacterial metabolic activity, the bacterial concentration is detected in real time by detecting the sensor potential, detailing its scientific significance, working principle, structural design and advantages. The results of detection of different concentrations of Escherichia coli in urine were in line with expectations. To make the article more complete scientifically sound, the following suggestions are put forward.

1. In the introduction part of the manuscript, only the social problems in urine bacteria were described, but not other social problems in bacteria detection. If the author can briefly introduce the social problems existing in the overall bacterial detection, the article will be more complete.

2. We suggest removing the superfluous horizontal lines at the bottom of Figure 3 (a) and comparing them neatly with Figure 3 (b). At the same time, correct the slight loss of the right most digit of the coordinate axis below Figure 4 to make the picture more neat

3. The author detected the Escherichia coli in urine, but did not explain the disease severity corresponding to different concentrations of Escherichia coli in urine. We suggest that the author explain this.

4. The manuscript indicates that the potential information can be received on the mobile phone, but we are puzzled whether the operator can accurately calculate the bacterial concentration through the potential information. Please briefly explain how the operator transforms the information.

5. In the introduction, some strategies of portable sensor are reviewed, while it seems some important categories are missing, please add and discuss, Such as Space-time-regulated imaging analyzer for smart coagulation diagnosis. Cell Rep. Med. 3, 100765 (2022) ; Touchable cell biophysics property recognition platforms enable multifunctional blood smart health care,” Microsyst. Nanoeng. 7, 103 (2021).

6. This technology uses bacterial metabolic activity to detect the overall bacterial concentration. We are interested in whether this equipment is expected to achieve uncalibrated detection of different bacteria in the future. Please brief introduce.

Reviewer 3 Report

This research is very interesting, but will be improved using a correct multivariable analysis to understand better variations in the results.

Is very important describe better this case of study to understand better the correct proposed solution.

Specific comments

In addition is important describe a Design of Experiments.

Explain with more detail the multivariable analysis.

Determine what is the correct way associated with the future research.

Using a more detail future research including the future limitations to this study.

Is very important using a comparative table with this data associated with this research.

Round 2

Reviewer 1 Report

In the revised version submitted this time, the author has made a targeted response to the last review comments. Personally, the version submitted this time has basically met the publication requirements of the journal. It is suggested to further improve the abstract and conclusion according to the main content of revised manuscript.

Author Response

We have updated the abstract as well as the conclusions to also include the new measurements we performed in different urine samples. We also added a sentence regarding planned future studies to the conclusions, to reflect what we added to the discussion part. Finally, we removed all the parts highlighted from the previous revision, and all changes are now shown in track-changes.